# Resurrecting Score11 in Siren:
# What ever happened to the 1980s score languages?

**Stephen Travis Pope**                                                    *stephen@heaveneverywhere.com*
*FASTLab and HeavenEverywhere, Ojai, California, USA*

**Reviewed on OpenReview:** *https: // openreview. net/ forum? id= jZcrFOwu12L*

## Abstract

This paper describes a "software archaeology" project in which a new interpreter was created for the *Score11* music representation, a popular 1980s music input language that was frequently used with the *Music11* non-real-time software sound synthesis package. The new version runs within the Smalltalk-based *Siren* system, a library of software classes for music representation, algorithmic composition and live interactive performance. The project background is given, and the port of Score11 to the Siren environment is described and evaluated.

## 1   Introduction

The first compositions of "computer music" involved the use of stochastic algorithmic composition programs to create music scores for performance by traditional instruments (Hiller & Isaacson, 1959). The development of what were called "music input languages" later progressed in parallel with the development of software sound synthesis (SWSS) languages in the 1960s and 1970s (Pope, 1993b). In fact, the seminal Music-V package (Mathews, 1969) incorporated a score preprocessing stage in which some form of algorithmic composition or other score manipulation could be applied to generate the note list before the audio synthesis begins (see (Mathews, 1969), p. 78ff).

The SCORE language is the work of Leland Smith, working at the CCRMA center at Stanford University (Smith, 1972), (Smith, 1980); it was used as a score-generation pre-processor for CCRMA's *Mus10* system. With the wide availability of the Music-11 system (Vercoe, 1978), Alexander Brinkman, working at the Eastman School of Music in Rochester, NY wrote a version of SCORE (with some revisions) that he called Score11 (Brinkman, 1981a), (Brinkman, 1981b).

This paper describes a project to create a new Score11 interpreter and integrate it into an interactive real-time system in Smalltalk , (Pope, 2002); the motivation was both to recreate several of the author's compositions from the early 1980s, and to enable new works to be created using the powerful Score11 description language.

## 2   Score Languages of the 1970s

Before MIDI and real-time software synthesis were available, computer music was created by non-real-time off-line software tools referred to as sound compilers or software sound synthesis (SWSS) packages (Pope, 1993b). In SWSS tools such as the "MusicN" family (for values of $N$ = I-V, 10, 11, 360, etc.), sound is computed by a non-real-time program that is configured by passing it two input files; one with the description of digital signal processing graphs that constitute the *instruments* of the orchestra; and another (the score or *notelist*) with a list of commands to play "notes" by activating the instrument definitions, passing them the parameters they require. The SWSS system then writes the output audio samples in sequence to the output sound file.

```
< This is a comment.
< Scorell Score for Terpsichore, Volte 201

* f 1 0 512 10 1;          < Generate a Sine in function 1
                           < * means in-line Music11 code
tempo 50 300;              < Tempo = 300 beats/min for 50 seconds

instrument 1 0 108;        < Instrument 1 for 108 beats
                           < p3 = Rhythm = 3 * 1/4, 2 * 1/8...
p3 rhythm 4/// 8// 4//// 2// 1/ 4/// 8// 4//// 2// 1/;
                           < p4 = Pitch = d5, e, fs, e...
p4 notes d5/e/fs/e/fs/g/fs/e/d/fs/g/a/
         d/e/fs/e/fs/g/fs/e/d/fs/e/d/;
p5 74;                     < p5 = Amplitude (in dB)
end;                       < end of instrument
```

Figure 1: Score11 example for a melody from Praetorius' *Terpsichore*

MusicN notelist files generally begin with a few lines that define the stored waveform and envelope functions used by the orchestra, followed by one or more sections of note commands, which are performed in sequence. There are standardized note command arguments for the required parameters—start time, duration, amplitude and probably pitch—while the parameters that define the note's other timbral/spatial characteristics follow them. The parameters of each note in the notelist are listed as an array of numbers that are referred to positionally (e.g., p5, p13) by the notelist reader. It is not unusual for instruments to have 20 or more parameters, so notelists for even short and simple musical selections can grow large and difficult to manage.

Starting with the Music-V package (Mathews, 1969), a score-preprocessing stage was defined as part of the SWSS process; several techniques arose for this, ranging from "music input languages" with which one could directly transcribe traditional Western music notation to more "programming-language-like" algorithmic composition tools. Researchers at the CCRMA center at Stanford University developed a dialect of MusicV called Mus10 (because it ran on a Digital Equipment Corp. [DEC] PDP-10 mainframe), and Leland Smith implemented the SCORE program for creating Mus10 notelists based on (relatively) readable descriptions of music material (Smith, 1972), (Smith, 1980); later in the 1980s, it was also used for high-quality music typesetting on personal computers.

The attraction of SCORE was that it allowed easy transcription of music from common-practice Western notation, as well as supporting all manner of serial, set-theoretic, stochastic and other compositional algorithms, and also provided several non-standard representations for time/duration, pitch, loudness and other common note parameters.

By 1980, the DEC PDP-11 mini-computer had become popular, and MusicV-style language called Music11 (Vercoe, 1978) was available from MIT (for an annual rental fee, which annoyed some users, since they were generally academics); Alexander Brinkman proceeded to develop his Score11 program (based loosely on Leland Smith's SCORE program) for use with Music11 (Brinkman, 1981a), (Brinkman, 1981b).

## 3   The Structure of a Score11 file

A musical score in Score11 consists of one or more text blocks that describe sequences of notes for a single instrument. The parameters of the instrument's notes are given on separate lines, generally starting with rhythms and pitches, and proceeding to the (possibly many) other parameters. Special keywords support the convenient specification esp. of rhythm and pitch values. An annotated example is given in Figure 1 (Pope, 1993a) (from a score written in 1982 for music first published in 1610).

Figure 1 is an excerpt from a Score11 score for a melody from "Terpsichore" by Michael Praetorius (shown in Figure 2, courtesy of the International Music Score Library Project, https://imslp.org). Score11 comments

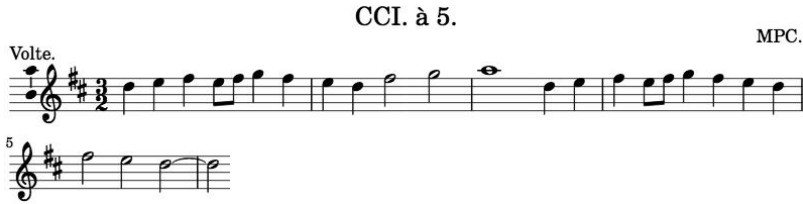

Figure 2: Common Western music notation for Figure 1

start with "<" and continue to the end of the line. The example's single instrument block starts with the *instrument* keyword and continues to the *end* keyword. Within the block, the statements define the data for the rhythm (p3), pitch (p4) and amplitude (p5) of the notes using different representations; the keyword *rhythm* introduces a list of note-duration values whereby a value of 4 denotes a quarter note and the number of '/' characters after the numerical value shows how many times the value is repeated. Similarly, the *notes* keyword allows one to use note names (with octave indications) for the p4 parameter to specify note pitch by name. A constant value is assigned once, as in the amplitude (p5) value of 74 for all notes.

In addition to the ability to enter note data directly as in this example, Score11 has a wide array of facilities for defining sets or sequences of values that can be repeated and transformed (e.g., a recurring phrase or a 12-tone row), and for defining static random selection ranges or dynamic "tendency masks" for values. A few examples of these options are given in Figure 3, and one rendition of the last pitch set in Figure 3 might look like the Hauer-Steffens notation shown in Figure 4.

## 4   The Port to Smalltalk and the Siren framework

The *Siren* system (Pope, 1992), (Pope, 2002) is a collection of software modules (classes) that encompasses:
(1) low-level music representation objects (duration, pitch, amplitude, timbre, spatialization, etc.),
(2) simple and composite musical events and event lists,
(3) objects that generate or modify event lists based on higher-level descriptions of compositional algorithms,
(4) real-time schedulers with which to perform event lists on their "instruments," and
(5) tools for building graphical user interfaces to interact with the other Siren objects.

A schematic view of these objects is given in Figure 5, which comes from (Pope, 2002). Given Siren's abstractions for event lists and event generators, it was straightforward to create a new subclass of event generator that parses an instrument block in a Score11 program and creates the corresponding Siren event list. The most important feature to keep from the Score11 syntax is the format of the parameter value lists shown in the examples above; it was decided that the "wrapper" text is relatively unimportant and could be easily changed to transform a Score11 instrument block into a legal Smalltalk object constructor expression.

Another alternative would be to use the Smalltalk compiler framework to build a parser for Score11 programs, but since the wrapper text is quite limited and simple, and Score11 programs tend to consist of many short blocks, the current approach was adopted.

Figure 6 shows the class inheritance hierarchy for the Score11 class in Siren/Smalltalk starting at the top with class *Object*; the tokens in parentheses after the class names are the given class' instance (member) variable names. One can see from Figure 6 that there are several levels of abstraction for the classes of musical events, event lists and event generators; specifically, event lists are themselves events that have collections of sub-events (an example of the object-oriented *composite* design pattern), and Score11 event generators are specialized event lists that have generator and postProcessor instance variables to facilitate their special event list creation methods.

The *Score11* class in Siren acts like a map or dictionary; instances are created with an instrument number and time-span as shown in the assignment to variable *s11* in Figure 7. Once the Score11 instance is created, parameter mappings can be added that associate a symbolic parameter name with an expression that

```
< Interpolation between values.
< Exponential ritardando over 20 beats (p3 doubles)
p3 movex  20   .1  .2;

< Crescendo for 5 beats (amplitude from 10 to 100),
< then Diminuendo for 5 beats
p5 move  5 10 100   /  5 100 10;

< Using sets of values.
< Choose at random for 10 beats from a C—major
< triad, then for 12 beats from a D—major one
p4 sets   10  c5 e g / 12   d5 fs a;

< Constant and random values.
< p8 will always be 100.3 (i.e., constant value)
p8 100.3;

< p9 will be between 3 and 8.5 100\% of the time
< (i.e., random selection range)
p9  1  3  8.5;

< p11 will be selected to be between 10 and 40 50\%
< of the time and between 40 and 45 50\% of the time
p11  .5 10 40   .5 40 45;

< Random selection ranges and moves combined
< to create tendency masks; notes chosen at
< random over 20 beats; starting range = c2—c3;
< final range is unison c5
p4  mo 20  c2 c3   c5 c5;
```

Figure 3: Score11 examples using alternative methods for specifying parameter values

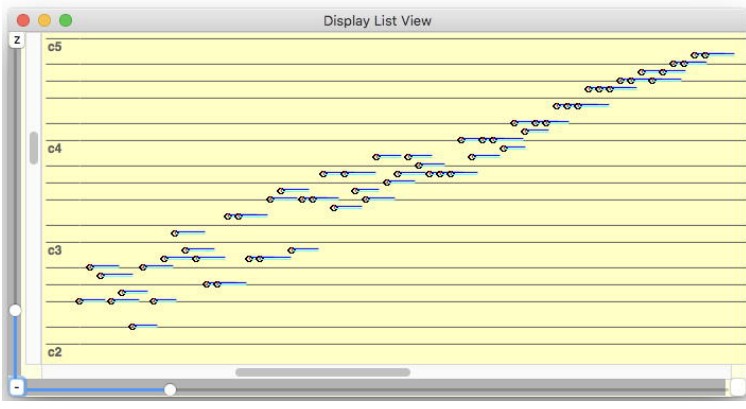

Figure 4: Hauer-Steffens notation for a structure with a dynamic tendency mask for pitch, as in the last expression of Fig. 3

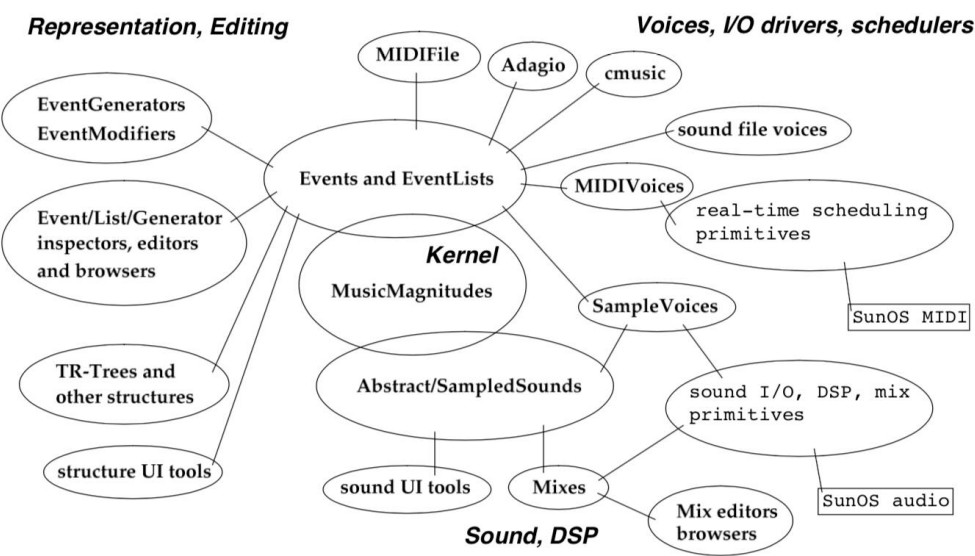

Figure 5: The components of the Siren library (from the documentation of an early-1990s version of MODE)

```
Object ()
    AbstractEvent (properties)
        DurationEvent (duration realTime)
            MusicEvent (pitch loudness voice)
                EventList (events index startedAt)
                    EventGenerator ()
                        Score11 (start stop instrument generators
                                    postProcessors paramMap)
```

Figure 6: Class inheritance hierarchy for the Score11 class in Siren/Smalltalk

```
chorale1
"Score for a Bach chorale — see
        http://scores.ccarh.org/bach/chorale/chorales.pdf #28"
"To test, execute [Score11 chorale1] and inspect the result."

| s11 elist |          "Declare variables"
                       "Instrument 1 block for 9 beats"
s11 := ((Score11 instr: 1 from: 0 to: 9)
                       "p3 = dur in beats; quarter and half notes"
    add: #p3 —> (#rh —> '4//////2//');
                       "p4 = chords; 'f3:a4' means 2 pitches at once"
    add: #p4 —> (#no —> 'f3:a4:c:f/f4:c5:f:a6/e4:c5:g:g/
                          eb4:c:f:a5/d4:d:f:bb5/c4:g:e5:c/f4:c:f5:a/');
                       "p5 = ampl ratio — random 0.25 to 0.3"
    add: #p5 —> #(1.0   0.25   0.3);
                       "Gliss ratio, constant 1.0 mapped to symbolic parameter name"
    add: #p6 —> 1.0 mapTo: #gliss:;
                       "L/R position is all over the map"
    add: #p7 —> #(1.0   1.0   —1.0) mapTo: #pos:;
                       "Modulation index is between 4 and 5"
    add: #p8 —> #(1.0   4.0   5.0) mapTo: #modInd:
).                     "End of constructor for s11"
s11 tempo: 100.        "Speed it up"
s11 du: 303.           "Duty cycle sets event dur to 3 x IOI"
                       "Generate the event list into the variable elist"
elist := s11 eventList.
```

Figure 7: Score11 example as a Smalltalk method

describes the parameter values using the keywords of Score11 scores. A trivial example of this would be the expression,

```
aScore add: #p6 —> 1.0   mapTo: #gliss:
```

which sets p6 to 1.0 and maps it to the *gliss* property.

The first few parameters have default mappings; p3, p4, and p5 are mapped to rhythm (which determines the event start/stop times and inter-onset intervals), pitch and amplitude respectively. For parameters beyond the basics, the *mapTo:* message is used to tell the interpreter which named property to assign based on the parameter data expression.

The example in Figure 7 shows a Bach chorale as notated in the new Score11 version; note that Smalltalk comments are enclosed in double quotes as in "This is a comment."

The *eventList* message sent to a Score11 object in the last line of the example creates and returns an event list based on the receiver object's maps of generators and postProcessors. In this process, the p3 parameter is first used to create a list of events with start times and durations only; after that, the other generators in the score's map are iterated over to add other properties to the events. Lastly, the postProcessors are used to transform, filter or otherwise manipulate the event list's items as described next.

The various Score11 representations for musical parameters (*rh, no, mo,* etc.) are implemented by simple generator methods that each iterate over the events in the event list and assign a named property (since Siren doesn't use positional parameters) using their policies, be they simple lists of values or complex generational algorithms. The methods for the handling of value sets and static or dynamic tendency masks within a parameter statement are actually quite simple. Figure 8 below shows the simplest example—the method that handles mapping constant-value parameters to the events in an event list.

Given an event list, Siren supports general-purpose post-processing using the message,

```
writeConst: property from: val into: eList
"Parse and process the constant−value keyword to
    generate events"
"E.g., aScore11 add: (#p5 −> 70);"

    | evts |                   "Declare a var name"
    evts := eList events.      "Get the list of evts"
    evts do:                   "Event loop"
       [ :eAss | | tN |        "Loop arg & temp"
                               "Get evt start time"
      tN := eAss time asSec value.
                               "If evt time in range, assign property"
      (tN >= start and: [tN <= stop]) ifTrue:
          [eAss event perform: property with: val]]
```

Figure 8: Example Score11 class generator method to process a constant-value parameter

```
anEventList applyBlock: aBlockOperation
            toProp: propName
            from: start
            to: stop
```

wherein the argument *aBlockOperation* is a function or closure that will be evaluated for each event in the list, as in,

```
anEventList applyBlock: [ :evt |
            (BohlenPierceScale root: PitchClass mi)
                nearestNoteTo: evt pitch asHz value]
        toProp: #pitch
        from: 24 to: 34.
```

which will round the eventList's note pitches between 24 and 34 beats to the nearest pitches on a Bohlen-Pierce scale rooted on E.

Among the motivations to port Score11 to Siren was the availability of the other Siren components: the additional event generator classes, the general-purpose Smalltalk programming language for manipulating event lists, and the GUI components that are bundled with Siren. The *applyBlock* example above shows the power of event list processing using standard Smalltalk data and control structures. These have proven very useful in developing extended musical fragments with the system. There are many more examples in the Siren source code (Pope, 2022), and several of these will be given in the presentation of this paper.

## 5  Discussion

Just as there are several more modern software sound synthesis packages than MusicV in common usage (Pope, 1993b), music input languages and general-purpose music representations are also still active areas of R&D. Mikel Kuehn's *nGen* (http://mikelkuehn.com/index.php/ng) is a modern Score11-like preprocessor used with Csound (https://csound.com); unfortunately nGen is not open-source. *CommonMusic* (http://commonmusic.sourceforge.net) and *SuperCollider* (McCartney, 2002) are both examples of a music input language built into a flexible programming language. Sadly, however, none of these systems combines the terseness of Score11 with the power and flexibility of a general-purpose programming language.

Andre Bartetzki's *CMask* (https://www.bartetzki.de/en/software.html) was a powerful front-end for Csound that used tendency masks similar to Score11's. It appears to no longer be maintained. Another project named "Score" is incorporated in Steven Yi's excellent *Blue* package for Csound (https://github.com/kunstmusik);

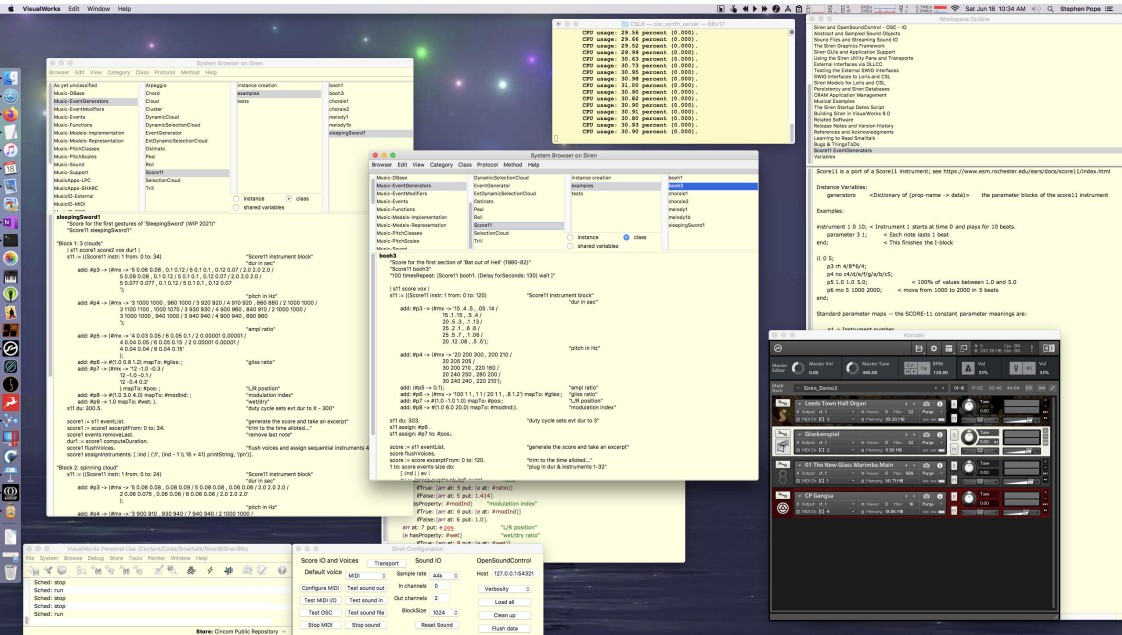

Figure 9: Score11 in Siren (cream-colored windows), with a yellow UNIX shell window running a CSL-based synthesis server that responds to OSC messages, and a black Kontakt MIDI sampler screen, both for real-time playback

it incorporates many of the features of Score11 together with a GUI for compositional objects and a Clojure-based extension mechanism.

The greatest drawback with the port presented here is the fact that Smalltalk has a small user base; nevertheless, there are new Smalltalk implementations and platform ports every year. It remains the author's tool of choice by virtue of its stability, portability and productivity, to say nothing of its syntactical simplicity and beauty, comprehensive class library, and integrated development environment.

Compared to the original tools with which Score11 was used (i.e., those found on UNIX on a PDP-11), the modern environment is both much more comfortable (e.g., the Smalltalk development tools), and supports real-time performance of scores due to Siren's support for both MIDI and OSC output (at the same time). The screen shot in Figure 9 shows an exemplary working setup; the yellow-tinted windows are VisualWorks Smalltalk, and one sees several kinds of tools: code browsers, editors, control panels and on-line documentation. The yellow window at the center-top is a UNIX shell window with a sound synthesis server running; it displays system usage messages. The black-background window at the lower-right is the GUI of Kontakt MIDI Sampler; both of these synthesis servers (OSC and MIDI) can be triggered by the same Siren event list and run in real time.

The Siren package is the latest in a line of Smalltalk-based frameworks for music and audio processing going back to the mid-1980s. The earliest version was called the HyperScore ToolKit and ran on Xerox Smalltalk-80 V2.3; this was then ported to the VisualWorks implementation and renamed the Musical Object Development Environment (MODE) around 1990. In the late-1990s, the entire system was ported to Squeak Smalltalk and renamed Siren. Siren was ported back to VisualWorks, which has been the main platform for almost 20 years. Currently a port to the CUIS Smalltalk implementation (https://cuis.st) is underway and will appear on the Siren github repository when available.

## 6    Conclusions

The new Score11 interpreter in the Siren/Smalltalk system has achieved the stated goals of (1) allowing the recreation of "ancient" (early 1980s) scores written for the original Score11/Music11 system, and (2) enabling the use of the algorithmic composition facilities of Score11 for new pieces (WIP). The fact that the resulting scores are simply Siren event list objects means that all manner of further processing, interaction and performance are made easy.

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
