# OpenReview forum: "Resurrecting Score11 in Siren: What ever happened to the 1980s score languages?"
_FAST.org.ar/2022/Workshop — FAST Smalltalk 2022_

### Official Review · Reviewer_KcUB · 2022-10-18
**Review of “Resurrecting Score11 in Siren: What ever happened to the 1980s score languages?”**

**Rating:** 7
**Confidence:** 5

**Review:**

The paper is nicely written and easy to follow, requiring only some familiarity with the topics discussed. It describes how the Siren music system (written by the author of the paper) was augmented with syntax to make it easy (although not automatic) to import scores originally written for the Score11 music system. I am convinced this is all original work by the author. With respect to relevance, truth is that the field of algorithmic composition has seen much evolution since the original inception of Score11 (70’s and 80’s) and Siren (80’s and 90’s). Relevance is limited to those still working with these methodologies, and those interested in the history of computer music (a relevant topic of study by itself). Even if the audience will be limited, it is a worthy read, and I recommend publication.

---

### Official Review · Reviewer_C4CE · 2022-10-29
**Acceptance**

**Rating:** 8
**Confidence:** 5

**Review:**

The paper describes an addition to the Siren music framework that allows the
user to manipulate data in the [Score-11](https://www.esm.rochester.edu/ears/docs/score11/index.html) format. *Score-11* was a preprocessor
program written by Aleck Brinkman to be used with *music11*, a non-real-time
audio synthesis and composition software that ran on the PDP-11 minicomputer.

By [introducing a new `Score11` class](https://github.com/stpope/Siren9/commit/0872b0d9123221c773a9f651eef297eab720971c), Siren allows an additional input syntax
that is both terse and expressive, inviting a timely collaboration between two
seemingly different ways of thinking about computer music, and of human-computer
interaction in general. It is difficult to separate ideology from the historical
constraints that permeate it. For a composer, having immediate aural feedback is
as seductive as a true siren song, and so it is no wonder that even old-school
software like Csound started in the last thirty years to gradually incorporate
more and more support for real-time synthesis and operation.

Besides those already included in the paper, similar efforts worth mentioning
are:

-   **[JMask](https://kunstmusik.github.io/blue-manual/users/reference/soundObjects/jmask/):** part of the Blue music environment, a graphically-interfaced  [CMask](https://github.com/kunstmusik/cmask)
-   **[Score](http://kunstmusik.github.io/score/):** which —besides choosing yet another “score” name— acknowledges being
    under the influence of Siren’s [SmOKe](http://kunstmusik.github.io/score/design.html)!

I’d welcome an opportunity to actually try the software but I was unable to
obtain a license for VisualWorks Smalltalk. Reproducible research is for me just
another name for the Dynabook. The irony of experiencing this impediment in the
context of Smalltalk is saddening. On a positive note, Siren’s source code is
freely available (although I couldn’t find a license note in its repository) and
the announcement of preliminary work towards a Cuis Smalltalk port is excellent
news.

Siren grew and evolved over many years out of its author’s necessities, it
cannot but eventually satisfy them. As if that were not enough, we are also
treated with the chance to use it, preserve it, and hopefully learn from it and
contribute back. It would be foolish to reject Mr Pope’s generous gift as would
be equally foolish not to commend this paper and foster the work it describes.